# Unicolored phosphor-sensitized fluorescence for efficient and stable blue OLEDs

Paul Heimel[1,2,3], Anirban Mondal [4], Falk May[3,5], Wolfgang Kowalsky[1,2], Christian Lennartz[1,3,6], Denis Andrienko [4] & Robert Lovrincic [1,2]

Improving lifetimes and efficiencies of blue organic light-emitting diodes is clearly a scientific challenge. Towards solving this challenge, we propose a unicolored phosphor-sensitized fluorescence approach, with phosphorescent and fluorescent emitters tailored to preserve the initial color of phosphorescence. Using this approach, we design an efficient sky-blue light-emitting diode with radiative decay times in the submicrosecond regime. By changing the concentration of fluorescent emitter, we show that the lifetime is proportional to the reduction of the radiative decay time and tune the operational stability to lifetimes of up to 320 h (80% decay, initial luminance of 1000 cd/m$^2$). Unicolored phosphor-sensitized fluorescence provides a clear path towards efficient and stable blue light-emitting diodes, helping to overcome the limitations of thermally activated delayed fluorescence.

[1] InnovationLab, 69115 Heidelberg, Germany. [2] Institute for High Frequency Technology, TU Braunschweig, 38106 Braunschweig, Germany. [3] BASF SE, 67056 Ludwigshafen, Germany. [4] Max Planck Institute for Polymer Research, 55128 Mainz, Germany. [5]Present address: Merck KGaA, 64293 Darmstadt, Germany. [6]Present address: trinamiX GmbH, 67063 Ludwigshafen, Germany. Correspondence and requests for materials should be addressed to D.A. (email: denis.andrienko@mpip-mainz.mpg.de) or to R.L. (email: r.lovrincic@tu-braunschweig.de)

Currently, external quantum efficiencies of deep blue organic light-emitting diodes (OLEDs) in commercially available display and lighting applications are limited by unfavorable spin statistics of fluorescent emitters, because operational lifetimes of (long desired) blue phosphorescent and thermally activated delayed fluorescence (TADF) OLEDs are too short to warrant industrial interest. OLED researchers have been exploring several routes trying to improve the stability and the efficiency of blue OLEDs[1–3]. Conceptually, harvesting excited triplet states boosts efficiency, whereas reducing the decay time of excited states decreases the probability of degradation reactions, triggered by triplet-triplet annihilation[4–6] and triplet-polaron quenching[7–9]. In practice, one has to find a compromise: phosphorescent emitters, for example, do harvest triplet states but have long excited state decay times. A different approach is to design fluorescent emitters capable of harvesting triplet excitons. Small singlet-triplet splitting of these emitters facilitates efficient (reverse) intersystem crossing, (R)ISC, from the triplet to the singlet state[10–12], leading to TADF[2,10–14]. Excited state decay times of TADF and phosphorescent emitters are, however, of the same order of magnitude, resulting in similarly fast operational degradation.

Stability and color purity of TADF OLEDs can be improved by combining TADF emitters with conventional fluorescent emitters in a sensitizing approach[15,16]. In such a hyperfluorescent TADF OLED, an efficient Förster resonance energy transfer (FRET) from the singlet excited TADF state to a singlet of the fluorescent emitter reduces the number of RISC/ISC cycles prior to radiative decay. The radiative decay time is, however, still limited by the rate of the RISC process: due to spin statistics, 75% of excited states need to be transferred to singlet states prior to FRET or fluorescence. Due to the quadratic dependence of the RISC rate on the spin-orbit coupling strength[17], the decay rate of the delayed emission is limited to $\sim 10^6\,\mathrm{s}^{-1}$[18,19]. The slow decay rate of the delayed emission, ultimately, limits the stability of hyperfluorescent TADF OLEDs.

An alternative to TADF approach, which also surpasses the 5–7 % theoretical limit for the external quantum efficiency of fluorescent OLEDs, is the phosphor-sensitized fluorescence approach, shown in Fig. 1(a)[20–24]. Generally speaking, FRET from a triplet to a singlet state is spin-forbidden and hence inefficient. Long decay time of the excited phosphorescent triplet state and a high photoluminescence quantum yield can, however, compensate for the slow FRET rate[25]. In principle, all excited states can be directly transferred to the fluorescent acceptor via FRET. Direct transfer results in shorter radiative decay times than TADF hyperfluorescence. This was demonstrated in a recent

work by Kim et al.,[26] showing a significant decay time reduction in the transient PL-characteristics of the well-known phosphorescent green-emitter Ir(ppy)$_3$ by adding a yellow emitting fluorescent acceptor. Sensitization is particularly interesting for white OLEDs, where a blue sensitizer is used to pump a yellow (two-component white OLED) or a green and red (three-component white OLED) emitter. A sensitized white OLED offers a stable white color with respect to changes in the driving voltage or degradation of blue sensitizer, since its emission depends exclusively on the exciton formation on the sensitizer[27–29].

In principle, an efficient fluorescent blue OLED could be realized using a sensitizing donor emitting in the UV spectral range. High exciton energies would, however, lead to a very fast degradation of the device. This is why the classic sensitization by a red-shifted (with respect to the donor) acceptor emission is not suitable for the realization of a stable blue OLED. Conceptually, a fluorescent acceptor with an emission spectrum matching the emission of the phosphorescent sensitizer, as schematically depicted in Fig. 1(b), can also be used. Thereby, the excitation energy required to pump the acceptor emission is effectively reduced. In fact, using acceptors with a narrow emission peak can even improve color purity. Until now, however, a number of practical design challenges prevented the realization of a unicolored sensitized emitting system. Due to large reorganization energies of organic emitters upon excitation (Stokes shift), the overlap of donor's emission and the acceptor's absorption is reduced in a unicolored phosphor-sensitized fluorescence (UPSF) system compared to a non-unicolored system. Indeed, assuming Gaussian shape of the peaks, we can estimate the spectral overlap $J$ to be proportional to $\exp\left[-\frac{\left(E_A^{\mathrm{abs}}-E_D^{\mathrm{em}}\right)^2}{2\left(\sigma_A^2+\sigma_D^2\right)}\right]$, where $E_A^{\mathrm{abs}}$ and $E_D^{\mathrm{em}}$ are the positions of the absorption and emission peaks of the donor and acceptor, as indicated in Fig. 1(b). In a unicolored system, $E_D^{\mathrm{em}}\sim E_A^{\mathrm{em}}$, hence $J\sim\exp\left[-\frac{E_{A,\mathrm{stokes}}^2}{2\left(\sigma_A^2+\sigma_D^2\right)}\right]$, i.e., is limited by the Stokes shift of the acceptor. Small FRET rates require higher concentrations of the acceptor. High acceptor concentrations facilitate Dexter transfer of excited triplets of the donor to the acceptor and, consequently, efficiency losses due to non-radiative quenching of excited states. The chemical design of UPSF OLEDs should therefore target donor-acceptor combinations with small Dexter and large FRET rates, at the same time optimizing the acceptor concentration to achieve a balance between OLED efficiency and lifetime.

In this work, we present the first realization of UPSF, an emitting system where the emission color of the sensitizer is

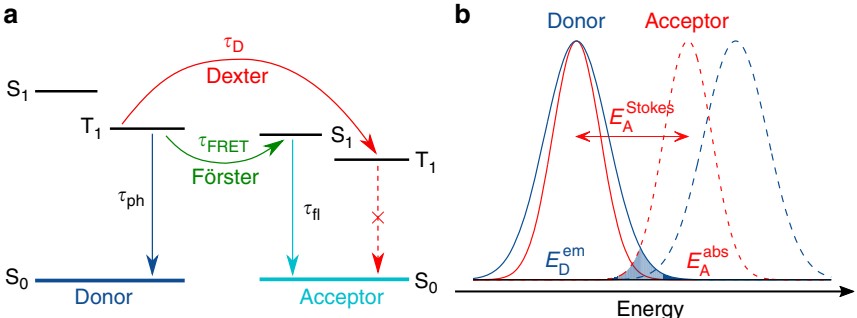

**Fig. 1** Working principle of unicolored phosphor-sensitized fluorescence. **a** Radiative decay paths of phosphor-sensitized fluorescence. Long-range Förster resonance energy transfer from the triplet state of the phosphorescent donor to the energy-matched singlet state of a fluorescent acceptor reduces excited state decay time while preserving the emission color. Unwanted Dexter transfer to the acceptor triplet state can lead to a reduced quantum efficiency. **b** Spectral properties: absorption (dashed lines) and emission (solid lines) of donor (blue) and acceptor (red) in a unicolored phosphor-sensitized fluorescence system. The shaded area indicates the spectral overlap $J$ of donor emission and acceptor absorption

preserved and the device stability increased. Using UV–visible as well as steady-state and time-resolved photoluminescence spectroscopies and molecular dynamics simulations, we elucidate photophysical properties of the UPSF system. In addition, we fabricate a series of bright, sky-blue UPSF OLEDs that show a linear relation between the radiative decay time reduction (by a factor of three) and OLED lifetime increase, without any emission color shifts. Our optimized devices are among the best published sky-blue OLEDs regarding a balance between efficiency and stability, while UPSF presents a clear path forward for long-term stable, highly efficient blue OLEDs.

## Results

**Förster energy transfer between donor and acceptor**. The chemical structures of phosphorescent donor (D)[30], fluorescent acceptor (A)[31], and matrix (M)[32] materials are shown in Fig. 2, together with the UV–vis absorption spectra of M-, D- and A-layers, and normalized PL spectra of both emitters in the matrix. The donor and acceptor emission spectra have, by design, similar spectral shapes and peak positions at 480 nm. Despite the narrower peak of the acceptor's emission, there is still some overlap between donor emission and acceptor absorption, which can facilitate Förster transfer of the excited donor states to singlets of acceptor.

To determine the Förster radius of the emitting system, we measured extinction spectra of acceptor dissolved in toluene at concentration of $10^{-6}$ M. The Förster radius of the donor, $R_{FRET} = 2.4$ nm, was estimated from the spectral overlap integral $J(\lambda)$ of the extinction spectrum of the acceptor and the normalized photoluminescence spectrum of the donor, which are shown in Supplementary Figure 1. The UPSF system has a smaller FRET radius compared to a non-unicolored system, which are typically in the range of 4–5.5 nm[20,33–35]. This reduction is due to a smaller spectral overlap of donor's emission and the acceptor's absorption.

**Radiative decay times**. To investigate FRET between donor and acceptor and its impact on the radiative decay times, we performed time-resolved PL (TRPL) measurements of mixtures and bilayers. To ensure the predominant excitation of the donor molecules, we used a 375 nm laser for excitation. This excitation wavelength is close to the minima of absorption of both matrix and acceptor (see Supplementary Figure 2) but is well within the maximum of the donor absorption. Using the corresponding absorption coefficients, we calculated that donor molecules absorb over a factor of 10 more light than acceptor molecules in all PL measurements, namely 95, 47, 32, and 10 times more for

0.5%, 1.0%, 1.5%, and 5% acceptor concentrations, respectively. The bilayer samples, sketched in Fig. 3, were composed of a donor and an acceptor layer separated by a thin spacer layer of matrix material. We used the following weight ratios: M:D(80:20, 10 nm)/$M_{spacer}$(0–6 nm)/M:A(95:5, 10 nm). The mixed samples were co-evaporated on glass substrates with a constant donor concentration of 20 vol %. We gradually increased the acceptor concentration from 0% (reference) to 1.5 vol %, at the same time decreasing the matrix concentration from 80% to 78.5%. Since the bilayer geometry excludes the Dexter transfer, we can use rather high acceptor concentration of 5% and improve the sensitivity of detection of FRET events. The emission was measured close to the peak maximum at 480 nm. Figure 3 shows the time-resolved spectra of the bilayer and mixed samples.

In the case of the bilayer, Fig. 3(a), a clear shift of slower events (>1 µs) to faster ones (<0.5 µs) is visible when the spacer thickness reaches values below 3 nm. This is due to an increase in FRET at donor-acceptor separations below the Förster radius of 2.4 nm. Because of the rapid FRET, less triplet excitons decay radiatively on the donor (slow phosphorescence), but undergo FRET and decay on the acceptor molecules (fast fluorescence). Hence, a net transition from slow to fast decays is observed.

For the mixed layers, Fig. 3(b), we performed multi-exponential fits to identify the average (intensity-weighted) fluorescence decay times, as described in the Supplementary Note 2: Fluorescence and radiative decay times. In order to extract the radiative decay time, we weighted these values by the corresponding quantum yields. The determined decay times of 1.60 µs, 1.06 µs, 0.77 µs, and 0.49 µs for 0%, 0.5%, 1%, and 1.5% acceptor concentrations, show that UPSF, indeed, leads to a reduction of the radiative decay time. The estimated values for the FRET-rates and transfer efficiencies are given in the Supplementary Note 3: Estimation of the Förster rate.

**Dexter energy transfer**. In addition to FRET to the singlet excited state of the acceptor molecules, Dexter transfer to the triplet states can occur in UPSF, which is detrimental to the efficiency of a UPSF system. To quantify the effect of Dexter transfer on the PL quantum yields (PLQY), we measured steady-state photoluminescence of 50 nm thick mixed M:D:A layers, which are shown in Supplementary Figure 3. Table 1 summarizes the PLQY values for all acceptor concentrations.

The sample with only donor shows the highest quantum yield of ~100%. Similarly, an emitter layer with acceptor only shows a high PLQY of 95%. For the mixed layers, the PLQY gradually decreases with the increase of the acceptor concentration (see Table 1), due to increasing Dexter transfer from the excited triplets of the donor to

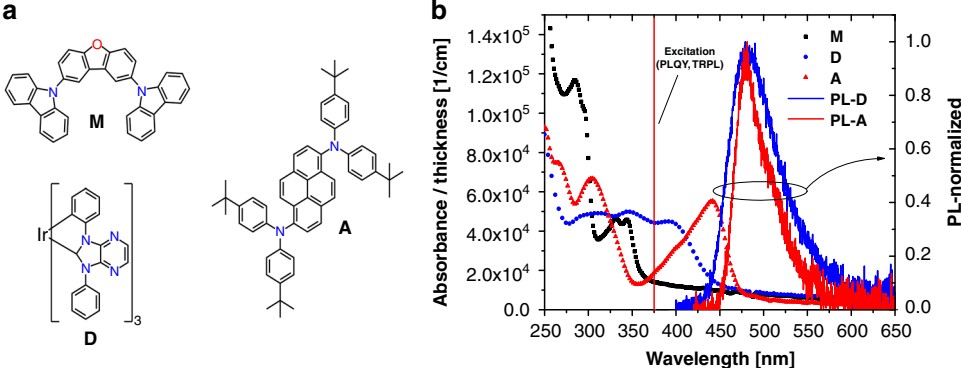

**Fig. 2** Molecular structures and photophysical properties of the unicolored phosphor-sensitized fluorescence system. **a** Molecular structures of the matrix, acceptor, and donor molecules. **b** UV–visible spectra of neat matrix, donor and acceptor layers on glass and normalized photoluminescence spectra of donor and acceptor in matrix. The red line shows the excitation wavelength of 375 nm used for the photoluminescent experiments

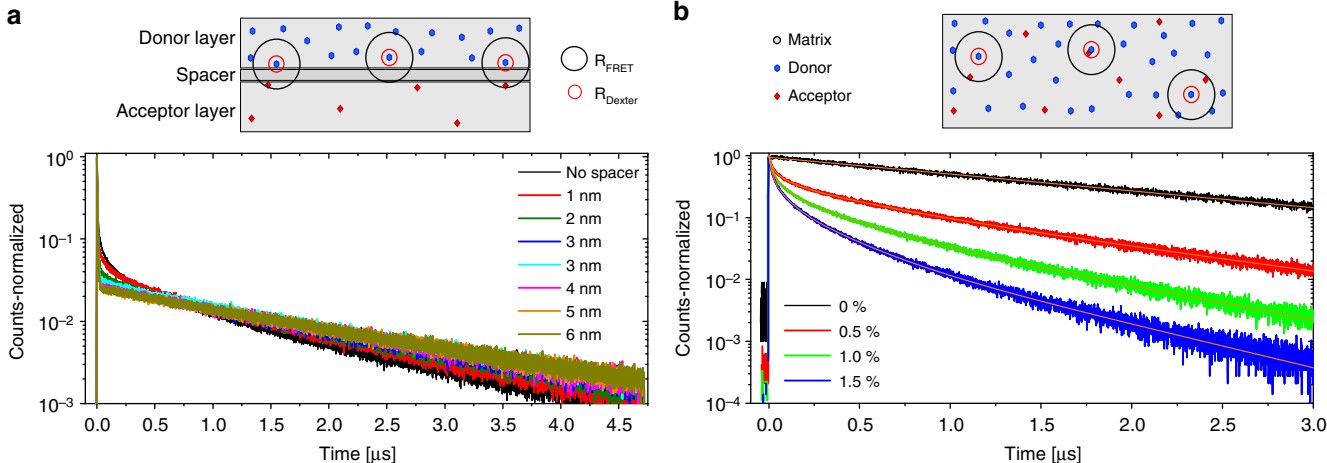

**Fig. 3** Time-resolved photoluminescent experiments. All samples were excited at 375 nm to suppress direct acceptor excitation. **a** Sketch and TRPL-data of bilayer samples. The thickness of the matrix spacer layer, which separates the donor from the acceptor layer, was varied from 0 to 6 nm. Förster-radius and Dexter-radius are depicted (not to scale) as black and red circles around the donor molecules. The inset shows a zoom-in to illustrate the increase of counts. **b** Sketch and TRPL-spectra of mixed layers. The A concentration was gradually increased from 0% to 1.5%. The decays were fitted with a multi-exponential fit (red lines) to derive the mean (intensity-weighted) radiative decay time

**Table 1 Photolumiescence quantum yields and decay times, external quantum efficiencies, color coordinates, and lifetimes for different donor/acceptor concentrations in the matrix**

| D:A [%] | 20:0 | 20:0.5 | 20:1.0 | 20:1.5 | 0:5/2 |
|---|---|---|---|---|---|
| PLQY | 100% | 82% | 66% | 63% | 95% |
| $\tau_{rad}$ (μs) | 1.60 | 1.06 | 0.77 | 0.49 | 0.004 |
| EQE(1000 nits) | 13.9% | 11.6% | 11.0% | 10.5% | 1.4% |
| EQE-relative | 100% | 83.5% | 79% | 75.5% | — |
| (CIEx,CIEy) | (0.156, 0.278) | (0.147, 0.275) | (0.142, 0.272) | (0.139, 0.274) | (0.133, 0.258) |
| LT$_{70}$ (25 mA/cm$^2$) | 26 h | 47 h | 60 h | 76 h | 16.8 h |
| LT$_{70}$ (4000 cd/m$^2$) | 46 h | 63 h | 71 h | 79 h | — |

The last column refers to systems without donor, with 5% acceptor for photoluminescent measurements and 2% for light-emitting diodes

the triplets of the fluorescent acceptor, where they are subsequently trapped and quenched[21,22,28,36–39]. At a donor to acceptor ratio of 20:1.5, already 25% of excitons undergo Dexter transfer from donor to acceptor. This highlights that the material system can be further optimized by reducing the Dexter transfer.

To get a better insight into the processes in mixed layers, we performed molecular dynamics simulations of their amorphous morphologies, as described in the Supplementary Note 6: Simulations. Using these morphologies (see Supplementary Figure 4), we calculated the effective FRET rate from a particular donor, to all the surrounding acceptor molecules. The distributions of these rates for all donors is rather broad, implying that we should not expect a monoexponential behavior of the FRET. This is indeed obvious from Fig. 3, where the intermediate timescale (up to a μs) is better described with a stretched exponential. The averaged, over all donors, FRET rate is shown in Supplementary Figure 5. Simulations predict that it saturates at ~3 vol %, at which point it is three times larger than $k_{ph}$. To study how the Dexter rate depends on the acceptor concentration, we have determined the number of neighboring acceptor molecules for every donor (see Supplementary Figure 6). It turns out that most donors have only one acceptor as a neighbor, hence the percentage of donor molecules participating in the Dexter transfer is proportional to the acceptor concentration.

The dependence of the FRET and Dexter rates on the acceptor concentration, tells us that, tuning the OLED operational lifetime and efficiency by changing the acceptor concentration is possible only within a certain range of acceptor concentrations (in our case up to 3%). Increasing the concentration further does not lead to a decrease of excitation decay times (and hence the operational stability of an OLED) but still increases quenching of triplet states via the Dexter transfer, thus reducing the overall OLED efficiency. To further improve OLED stability and efficiency, one has to chemically design donors and acceptors which favor long-range energy transfer and disfavor short-range migration of triplet states. In fact, a recent study showed that by attaching 2-phenylpropan-2-yl spacer ligands to an acceptor molecule, and by thoroughly engineering of an OLED device, quenching due to Dexter transfer can be suppressed in a TADF-sensitized OLED, allowing for acceptor concentrations as high as 3–5 weight %[40]. This shows that a UPSF-system with a maximum FRET rate at 3 volume % and without the drawbacks of reduced quantum efficiency from Dexter transfer is indeed feasible. Our estimate predicts that, by suppressing the Dexter transfer, the FRET rate can be increased by at least a factor of two (see Supplementary Figure 7), which would result in a total increase of the radiative decay rate by a factor of seven, since $k_{rad} = k_{ph} + k_{FRET} = k_{ph} + 6k_{ph} = 7k_{ph}$.

**OLED characteristics.** We fabricated a series of UPSF-OLEDs with acceptor-doped emission layers, to demonstrate the feasibility of our UPSF approach. The full stack information, including the chemical structure of all compounds used in the OLED, are shown in Supplementary Figure 8. The current–voltage ($J$–$V$) characteristics of the UPSF OLEDs do not change with acceptor concentration, implying that the acceptor molecules do not participate in charge transport. The electrons are transported only by the donor-molecules, which ensures exciton formation exclusively on donors (see Supplementary Note 7: OLED stack and materials for a discussion of charge transport in the UPSF OLEDs). Subsequently, excitons either decay radiatively on donor molecules by phosphorescence, or transfer to the acceptor molecules via FRET, and decay fluorescently.

EQE versus luminance, Fig. 4(b), shows a decrease in EQE with increasing A concentration. At a luminance of 1000 cd/m² we extracted values of 13.9, 11.6, 11.0, and 10.5%. Note that the moderate EQE value of the D-only device is due to isotropic orientation of the emitter in the film and can be further increased by optimizing the emitter orientation, which reduces light out-coupling losses[41,42].

In Table 1 the relative EQE values are compared to the previously determined PLQY values. Considering a potential deviation of +/−0.1% in the respective doping concentrations, as well as an error of +/−5% for the PLQY values, the EQEs are in a good agreement with the PLQY values. We conclude that, the decrease in EQE has the same origin as the decrease in PLQY, namely exciton trapping on the acceptor triplets due to Dexter transfer from the donor triplet states.

In Fig. 4c, d we compare the normalized EL and PL spectra of the mixed layers and corresponding OLEDs. The spectra of matrix:acceptor (acceptor-only) samples are included for reference. In both cases, a narrowing of the emission peak width can be observed with increasing acceptor concentration, gradually approaching the spectral shape of the acceptor-only sample. This proves that a part of the luminescence stems from acceptor molecules. Since we ensured the predominant excitation of donor by carefully choosing the excitation wavelength and by energy-level management of the OLEDs (EL-spectra), we conclude that

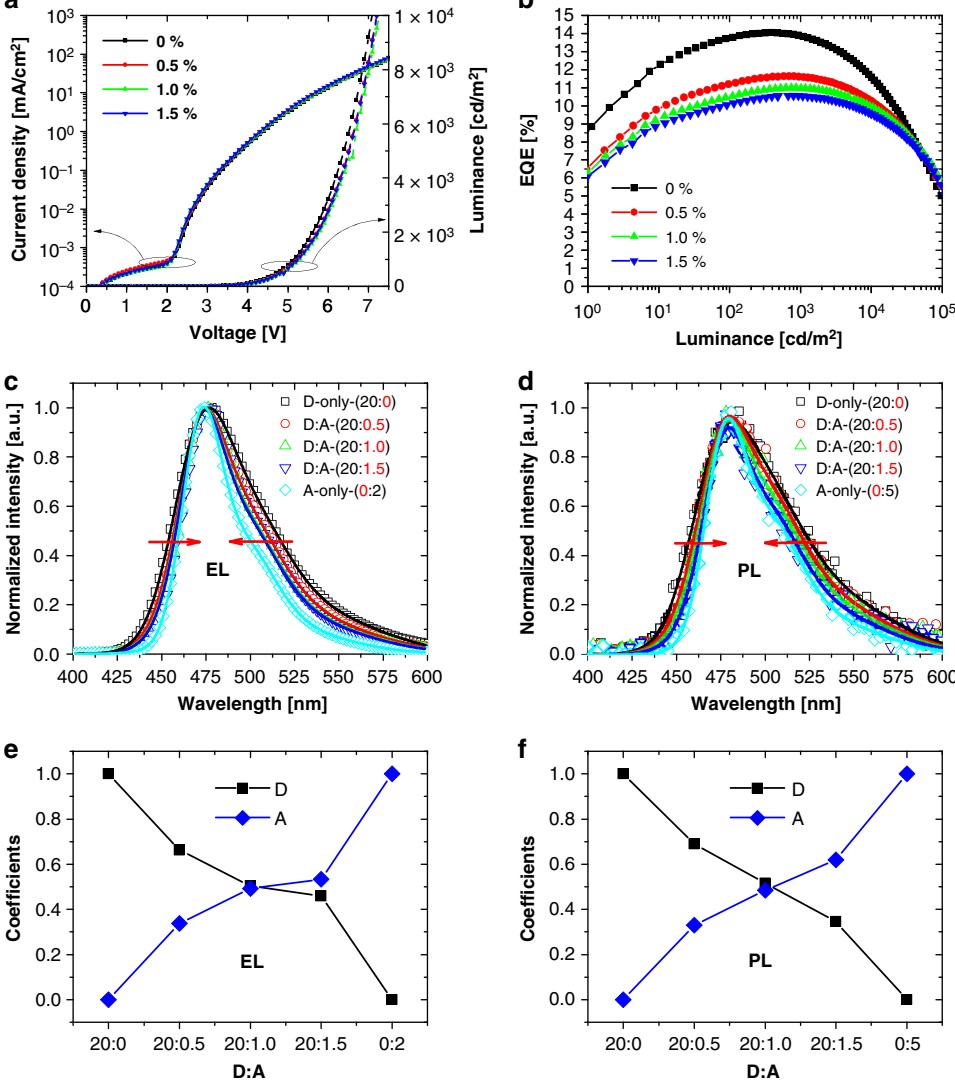

**Fig. 4** Characteristics of the light-emitting diodes (OLEDs) using mixed emission layers. The acceptor concentration was varied from 0 to 1.5%. **a** J-V-L characteristics. **b** External quantum efficiency vs luminance. **c** Normalized electroluminescence spectra and **d** normalized photoluminescence spectra. Donor-only and acceptor-only photoluminescence spectra were fitted with three Gaussian peaks. The mixed samples were fitted with linear combinations of the donor-only and acceptor-only fits: **e** and **d** show the donor and acceptor coefficients with respect to the acceptor concentration

the observed fluorescence of acceptors is due to FRET from the excited donor molecules.

A comparison of the corresponding CIE-coordinates in Table 1 (see also Supplementary Figure 9) shows that the emission color of the phosphorescent sensitizer is indeed preserved. Notably, this color can even be tuned towards a deeper blue by using an acceptor with a specific spectral shape, as shown in Supplementary Figure 10. To estimate the ratio of phosphorescent to fluorescent emission, we fitted the normalized spectra of the donor-only (M:D) and acceptor-only (M:A) samples with three Gaussians, and used their linear combination to reproduce the spectra of the M:D:A mixtures, which is shown in Fig. 4c, d. The respective coefficients for phosphorescence (D) and fluorescence (A) are given in the insets. Both PL and EL, show a clear shift of phosphorescence to fluorescence with increasing acceptor concentration, reflecting the increase in the FRET rate (see Supplementary Figure 11 for a comparison to the estimated transfer efficiencies of the FRET).

UPSF OLED $LT_{70}$ lifetimes, that is the period over which the luminance decreases to 70% of its initial value, were measured at a constant current. To compare two different stress conditions, we measured $LT_{70}$ at the same initial current density of $25\,mA/cm^2$ $\left(LT_{70}^{25}\right)$ and at the same initial luminance of $4000\,cd/cm^2$ $\left(LT_{70}^{4000}\right)$. The corresponding decay profiles are shown in Supplementary Figure 12. Figure 5 shows $LT_{70}^{4000}$, $LT_{70}^{25}$ as well as the inverse of the radiative decay time as a function of the acceptor concentration. For $LT_{70}^{25}$, in addition to a remarkable (factor of three) increase in the OLED lifetime for 1.5% acceptor concentration, we also observe an excellent correlation between the relative increase of the decay rate and the increase of the device lifetime. The $LT_{70}$ values are summarized in Table 1. For $LT_{70}^{4000}$, the decrease in current-efficiency has an additional (detrimental) impact on the OLED lifetime explaining its smaller increase with the acceptor concentration. Indeed, quenching of excited donor states by Dexter transfer to the acceptor, requires a higher current to achieve an initial luminance of $4000\,cd/cm^2$. In addition, trapped on the acceptor, triplet excitons contribute to the overall population of long-living excited states, leading to a faster degradation of the device.

To compare the lifetime of the OLED with 1.5% acceptor, measured at the initial luminance of $4000\,cd/m^2$, with the most stable published blue phosphorescent OLEDs[43,44], we also estimated $LT_{80}$ at $1000\,cd/m^2$ (see the Supplementary Note 12: Lifetime estimation). This estimate yields $LT_{80}$ ~ 320 h, which is close to the highest published value of $LT_{80}$ = 334 h for a blue phosphorescent OLED[44]. Our UPSF OLED has, however, a higher EQE (10.4% vs 9.6%) and deeper blue emission ($CIE_y$ = 0.27 vs $CIE_y$ = 0.30). Furthermore, the device stability of our UPSF system already compares favorably to recently published sky-blue TADF OLEDs[45] with $LT_{80}$ of 94 h measured at $500\,cd/m^2$. A conservative estimate of the lifetime of a UPSF-system with suppressed Dexter transfer predicts a lifetime boost by a factor of seven, exceeding $LT_{80}$ ~ 1200 h at $1000\,cd/m^2$.

## Discussion

We have proposed a triplet-singlet dual emitting system based on unicolored phosphor-sensitized fluorescence (UPSF). Steady-state and TRPL spectroscopies of model emission layers demonstrated that UPSF reduces the radiative decay time of the donor. The reduction is due to the fast resonant energy transfer of excited triplets to singlets of the fluorescent acceptor. Fabricated blue UPSF OLEDs preserved the original emission color of the donor. The increase in their lifetime, directly proportional to the inverse of the radiative decay time, opens up an optimization route for efficient and stable blue OLEDs. We foresee that the full potential of UPSF can be unlocked via (known) chemical design strategies. In particular, fast resonant energy transfer can be boosted by co-orienting the emitters. Likewise, Dexter transfer can be reduced by shielding their localized excited state wavefunctions by appropriate spacer groups. Finally, acceptor molecules with smaller Stokes shift will enable the fabrication of stable deep blue OLEDS, which meet the requirements for display applications. Our results show a clear path toward highly stable and efficient blue OLEDs.

## Methods

**Sample fabrication**. All samples were prepared by thermal vacuum deposition in the UHV ($\leq 10^{-7}$ mbar). Doped samples were fabricated by co-evaporation of the materials, while controlling each deposition rate independently with a respective microbalance. The photoexcited samples were prepared on glass substrates (Borofloat 33, Schott) that were precleaned by acetone and isopropanol treatment in an ultrasonic bath. The single OLED layers were deposited sequentially, without breaking the vacuum, onto glass substrates patterned with a 120 nm thick indium tin oxide (ITO) layer. Prior to the deposition, the substrates were cleaned by rinsing with isopropanol and subsequent ozone treatment for 20 min. The final devices were encapsulated with a glass cover in nitrogen atmosphere using an UV-epoxy resin.

**Photophysical measurements**. UV–vis measurements were performed in transmission mode using an AvaLight-DHS-Bal (Balanced Deuterium-Halogen light source (200−2000 nm)). Extinction spectra of B dissolved in toluene were recorded on a jasco UV/Vis V-670 spectrophotometer. Photoluminescence spectra including the derived quantum yields were measured under current nitrogen flow using a PTI QuantaMaster 40 spectrofluorometer equipped with an integrated sphere (Labsphere) and a Hamamatsu R928P photomultiplier. The samples were excited at wavelengths of 355, 375, and 410 nm with a Xenon lamp using a monochromator. The TRPL spectra were recorded under current nitrogen flow using a time-correlated single photon counting (TCSPC) system (Horiba Fluorocube). TRPL samples were excited at 375 nm with a pulsed laser diode (Horiba NanoLED Model N-375L).

**Device characterization**. J-V-L characteristics were measured using a sourcemeter (Keithley 2400) and a luminance meter (Minolta LS-100). OLED lifetimes were recorded at room temperature.

**Molecular dynamics simulations**. All Lennard-Jones parameters are taken from the OPLS force field[46–49], including the combination rules and a fudge-factor for 1–4 interactions of 0.5. Missing improper and torsional potentials are reparametrized based on QM scans[50]. Atomic partial charges of the acceptor, host, and donor molecules are computed by using density functional theory (wb97xd functional and 6–311+g(d,p) basis set), except for the Ir atom in the donor molecule for which pseudopotential equivalent to the def2-tzvp basis set is used. Ground state optimization of isolated molecules are performed and minimized configurations are further used to compute the electrostatic potential at the same level of theory and subsequently, atomic partial charges via the CHELPG[51] scheme. Molecular

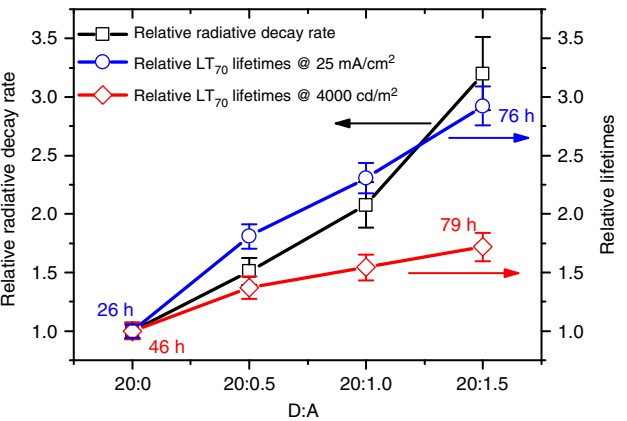

**Fig. 5** Correlation between lifetimes and radiative decay rates. Comparison of the relative radiative decay rates (black) and relative lifetime values (70% decay, $LT_{70}$) as a function of acceptor concentration. The device lifetimes were measured at constant current with initial current density of $25\,mA/cm^2$ (blue) and initial luminance of $4000\,cd/m^2$ (red)

dynamics simulations were performed using the GROMACS package. Mixtures were first equilibrated above their glass transition temperature and then quenched to room temperature.

## Data availability

Force fields and simulation data are available on request from the authors.

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

## Acknowledgements

This project has received funding from the BMBF grant InterPhase (FKZ 13N13661, FKZ 13N13656) and the European Union Horizon 2020 research and innovation program "Widening materials models" under Grant Agreement No. 646259 (MOSTOPHOS).

## Author contributions

C.L. devised the original idea for the project. P.H. and R.L. planned the experiments and analyzed the experimental data. P.H. performed all experiments. A.M., F.M., and D.A. performed molecular dynamics simulations and analyzed the results. P.H., D.A., and R.L. wrote the manuscript with input from all authors.

## Additional information

**Competing interests:** The authors declare no competing interests.

