## [Peer Review File · Nature Communications]

Reviewers' comments:

Reviewer #3 (Remarks to the Author):

I reviewed the manuscript "Unicolored phosphor-sensitized fluorescence for efficient and stable blue OLEDs". In this work the authors have developed phosphor sensitized blue LEDs. This area of research is important since the development of commercially demanding efficient blue materials and devices is still lagging behind other LEDs. If this concept would be successful and if large area devices could be fabricated using this method, I believe that this would be an important contribution not just in the area of blue LED but for White light generation as well.

I found the concept of utilizing phosphor-sensitized fluorescence approach in this work by the authors very appealing as compared to other mechanistic approaches for generating blue LED devices. However, before I recommend it for publication, I have the following major concerns on this work that has to be incorporated with appropriate experiments and correlation with the explanations.

Primarily, what was the basis for the design of BA, BD and BM? The overlap between BD emission and BA absorption is very less. Despite knowing this experimentally why were other molecules with more overlap not chosen? As a general observation authors have chosen phosphorescent emitter (BD) as donor and fluorescent (BA) as an acceptor for large FRET but the need for another fluorescence matrix incorporation and the concentration (?) of matrix remains unclear.

In continuation of the above, the authors have incorporated additional phosphorescent (HTM) and fluorescent (ETM) material (apart from BA and BD) in the UPSF-OLED. What is the necessity and physical significance of these extra layers and their role in the mechanism of pure blue light generation is not clear. What is the efficiency of the devices if these additional layers are not incorporated?

Overall in the reported UPSF-OLED-stack devices, there are several supporting layers as part of the device architecture deposited by various techniques. Hence, I do not think that this is a cost-effective device and its feasibility to fabricate over a flexible substrate is diminished. Any attempt to commercialize this architecture device will have this as a major drawback.

I did not find any mention about the theoretical/experimental ΔE_{ST} values for the chosen fluorescent emitter. This has to be provided.

Regarding the data of the devices I have the following observations:

Authors have not presented any morphological (AFM/FESEM) studies or the correlation of this Donor acceptor system in the matrix or without matrix.

I suggest to perform standard morphological analysis such as GIWAXS/GISAXS for each of the layer of the device and correlate their effect with the performance of the device. It is also necessary to provide the roughness data of the layers.

The authors are suggested to provide the voltage dependence EL spectra of the devices as a proof for the stable blue light.

Reviewer #4 (Remarks to the Author):

In this manuscript, Heimel et al. describe the a blue "unicolored" OLED operating via FRET from a phosphorescent donor to a fluorescent acceptor/emitter. While a similar mechanism is known broad band OLEDs, this is the first realization of a unicolor OLED operating by this mechanism. In contrast to previous manifestations, single color FRET device is particularly challenging because it requires strong similarity in emission profiles between donor and acceptor, good spectral overlap

between the donor emission and acceptor absorption, and fast and efficient FRET relative to the non-productive Dexter mechanism. The authors have selected and/or designed the materials as to meet these criteria and have provide strong experimental support for a FRET based mechanism using both emission and electroluminescent measurements. They also demonstrate that the operational lifetime of the device is increased due to the shortened excited state lifetime of the emissive layers. The increased device operation lifetime does come at the cost of a lower EQE but it is a solid proof of concept for this class of devices and the EQE could be improved. The manuscript is thorough, well written, and has high potential to impact both OLED science and application, so therefore the manuscript merits publication in Nature Communications after the minor revisions noted below.

The word lifetime is used throughout the manuscript. Since it can refer to both the excited state lifetime and device operational lifetime, it may be worth clarifying which is being referred to.

“no efficient blue emitters are commercially available” I don’t understand this statement. There are a number of OLED TVs and phones screens that contain blue emitters. What are the authors referring to?

Page 1: “large excited state lifetimes” to “long excited state lifetimes”

Figure 2a: Out of curiosity, why BM, BD and BA and not just M, D, and A? Including the B unnecessarily doubles the length of the label and increases their similarity at quick glance. Interestingly, the simulations section of the SI uses only M, D, and A.

Figure 2b: The absorption shapes are so closely spaced that the shapes are indistinguishable.
Figure 2b: How were the molecules deposited on the glass?

Figure 3: How was the Dexter radii determined?

Page 6, paragraph 2: “singlet of the acceptor” to “ singlet excited state of the acceptor”

The simulations assume random orientation ($k_2 = 2/3$) but the calculated J uses $k_2 = 0.476$. Why the discrepancy?

Figure 4c and 4d: For clarity the y-axis should be labeled Normalized Intensity.

The order of the SI does not match the order in the text. For the sake of the reader, the organization should coincide.

What is BA2 from Figure S4?

Figure S10: For those who print in black and white it would be useful to change the shape of one of the circles.

It would be helpful to see the coordinates in the CIE diagram in the SI.

Reviewer comments (black), our responses (blue), and changes in the manuscript (red)

Reviewer #3

Primarily, what was the basis for the design of BA, BD and BM? The overlap between BD emission and BA absorption is very less. Despite knowing this experimentally why were other molecules with more overlap not chosen?

Increasing the overlap between the donor emission and the acceptor absorption is indeed the main challenge of UPSF. This is why one of the key results of this work is that it is actually possible to realize energy transfer in a unicolored system. In the introduction we do mention that the overlap is limited by the Stokes shift (reorganization energy) of the acceptor. By decreasing this energy (stiffening the molecule) one can improve the overlap. In conventional sensitizing, it is possible to increase the overlap by pushing the pump further into the deep blue/UV. The issue with this approach is a very fast degradation of the pure deep blue phosphorescent OLED: it is practically impossible to remedy this degradation by doping with a fluorescent emitter. This is now explained in the manuscript.

*“In principle, an efficient fluorescent blue OLED can be realized using the sensitizing donor emitting in the UV spectral range. High exciton energies would, however, lead to an extremely fast degradation of the device. This is why the classic sensitization with a red-shifted (with respect to the donor) acceptor emission is not suitable for the realization of a stable blue OLED. Conceptually, a fluorescent acceptor with an emission spectrum matching the emission of the phosphorescent sensitizer, as schematically depicted in **Error! Reference source not found.**(b), can also be used. Thereby, the excitation energy required to pump the acceptor emission is effectively reduced. In fact, using acceptors with a narrow emission peak can even improve color purity. However, a number of practical design challenges prevented the realization of a unicolored sensitized emitting system so far.”*

As a general observation authors have chosen phosphorescent emitter (BD) as donor and fluorescent (BA) as an acceptor for large FRET but the need for another fluorescence matrix incorporation and the concentration (?) of matrix remains unclear. In continuation of the above, the authors have incorporated additional phosphorescent (HTM) and fluorescent (ETM) material (apart from BA and BD) in the UPSF-OLED. What is the necessity and physical significance of these extra layers and their role in the mechanism of pure blue light generation is not clear. What is the efficiency of the devices if these additional layers are not incorporated?

The matrix BM is a typical matrix used in phosphorescent blue OLEDs. It prevents self-quenching of emitter molecules and does not participate in fluorescence. The additional phosphorescent matrix material reduces the polaron stress of the emitter. It participates only in charge transport, not emission.

In general, the requirements for the materials/layers are fairly straightforward: All materials in the emissive layers have sufficiently high triplet energies in order to avoid quenching of triplet excitons. The fluorescent emitter BA does not participate in charge transport: its non-

radiative triplet recombination would reduce the device efficiency. This can be seen from, for example, Figure 4a. Here, the I-V curve is not changing when BA is added, indicating that it indeed does not participate in charge transport. The concentration of the matrix material is provided for all experiments.

To clarify the above, we have added the following paragraph to the SI detailing the working principle of our OLED:

“The UPSF-OLEDs are based on a state-of-the-art OLED stack (see **Figure S 8** for the OLED stack and materials used). Under operation, charge carriers are injected into the organic semiconducting materials by the aluminum cathode (electrons) and the transparent ITO anode (holes). For improved injection and charge transport, doped transport materials for both electrons (ETL) and holes (HTL) and an additional injection layer on the electron side (EIL) were used. To prevent charge carrier leakage from the emission layer (EML), HTM and M were used as electron blocking (EBL) and hole blocking (HBL) layers, respectively. Thereby, exciton formation is restricted to the EML. Furthermore, both HTM and M possess higher triplet energies to avoid energy transfer to those materials (exciton quenching). The emitter molecules are embedded in a matrix M in order to avoid self-quenching and reduce polaron-stress on the emitter. To reduce the number of holes conducted by the emitter molecules and restrict exciton formation to the donor molecules, we used an additional hole transport material in the EML. As a result, the amount of the matrix material was slightly reduced, with an overall doping ratio in the EML of $70-c_A:20:10:c_A$ for M:D:HTM:A, where c_A is the doping concentration of the acceptor. This can be explained by the higher doping concentration of the HTM and its large electric dipole moment of 8 Debye,⁵⁴ which supports trapping of the holes on HTM.⁵⁵”

Overall in the reported UPSF-OLED-stack devices, there are several supporting layers as part of the device architecture deposited by various techniques. Hence, I do not think that this is a cost-effective device and its feasibility to fabricate over a flexible substrate is diminished. Any attempt to commercialize this architecture device will have this as a major drawback.

Commercially available OLEDs have a comparable number of individual layers. Here, all layers are deposited by the same technique, namely evaporation in UHV, which is currently used in all commercial OLEDs.

I did not find any mention about the theoretical/experimental ΔEST values for the chosen fluorescent emitter. This has to be provided.

This is a typical fluorescent emitter with singlet-triplet splitting $> 0.1\text{eV}$. ΔEST is only relevant for TADF emitters.

Regarding the data of the devices I have the following observations: Authors have not presented any morphological (AFM/FESEM) studies or the correlation of this Donor acceptor system in the matrix or without matrix. I suggest to perform standard morphological analysis

such as GIWAXS/GISAXS for each of the layer of the device and correlate their effect with the performance of the device. It is also necessary to provide the roughness data of the layers.

OLED materials in high-efficiency vacuum-processed OLEDs are designed to form amorphous layers, because crystallization normally leads to rough films and device failure, see e.g.,

<https://www.sciencedirect.com/science/article/pii/S0009261405014727>

In amorphous materials, diffraction-based methods such as GIWAXS, will not yield useful morphological information. Similarly, since OLED layers themselves are very smooth, AFM will only measure the roughness of the ITO substrate.

The authors are suggested to provide the voltage dependence EL spectra of the devices as a proof for the stable blue light.

Please find the measurement below, which proves the stability of the color coordinate.

We also estimated the FRET efficiencies and compared them to the proportion of acceptor emission to highlight that the acceptor's emission is indeed pumped by the FRET of excited donor states, and hence independent on changes in driving voltage. To clarify this, we have modified the main text accordingly:

“The estimated values for the FRET-rates and transfer efficiencies are given in the Supplementary Information.”

“(see **Figure S 11** of the Supporting Information for a comparison to the estimated transfer efficiencies of the FRET).”

Which now refers to the following sections in the SI.

Estimation of the FRET-rates and transfer efficiencies

The FRET-rates k_{FRET} of the mixed layers were estimated from the radiative decay rates according to the following approximation:

$$k_r = \tau_r^{-1} \approx k_{ph} + k_{FRET}$$

where k_{ph} is the radiative decay rate of the donor in absence of the acceptor.

The transfer efficiencies E_{FRET} , giving the proportion of excitons decaying radiatively due to FRET to the acceptor, are given by:

$$E_{FRET} = \frac{k_{FRET}}{k_{ph} + k_{FRET}}$$

The calculated values are summarized in the table below.

BA (%)	0.5	1.0	1.5
k_{FRET} [10^6 s^{-1}]	0.32	0.67	1.42
E_{FRET}	0.34	0.52	0.69

“Comparison of the PL-coefficients and FRET-efficiencies

As written in the main text, we concluded that the observed fluorescence of acceptors in the EL- and PL-spectra (see **Figure 4** (c) (d)) is due to FRET from the excited donor molecules. To strengthen this conclusion we compared the estimated FRET efficiencies E_{FRET} for the mixed layers to the coefficients D (phosphorescence) and A (fluorescence) from the PL-spectra fits in **Figure S 11**. The trend of E_{FRET} closely follows the increase in A, reflecting the proportion of fluorescence, with increasing acceptor concentration. This indicates, that the fluorescence is indeed a direct result of FRET from excited donor states.

Figure S 11 Comparison of PL-coefficients and FRET efficiencies. The D-only and A-only PL spectra (Figure 4(c, d)) were fitted with three Gaussian peaks. The mixed samples were fitted with linear combinations of the D-only and A-only fits. The PL-coefficients (D and A), reflecting the proportion of phosphorescence and fluorescence, are plotted depending on the BA concentration. Additionally, the estimated FRET-efficiencies E_{FRET} are plotted.

Reviewer #4

The word lifetime is used throughout the manuscript. Since it can refer to both the excited state lifetime and device operational lifetime, it may be worth clarifying which is being referred to.

When referring to the lifetime of the excited state we replaced “lifetime” by “decay time”.

“no efficient blue emitters are commercially available” I don’t understand this statement. There are a number of OLED TVs and phones screens that contain blue emitters. What are the authors referring to?

To clarify our point we have changed the text to:

“Currently, external quantum efficiencies in state-of the art deep blue OLEDs in use for commercially available display and lighting applications are limited by unfavorable spin statistics of fluorescent emitters, as the operational stability of (long desired) blue phosphorescent and TADF OLEDs is insufficient.”

Page 1: “large excited state lifetimes” to “long excited state lifetimes”

This has been fixed.

Figure 2a: Out of curiosity, why BM, BD and BA and not just M, D, and A? Including the B unnecessarily doubles the length of the label and increases their similarity at quick glance. Interestingly, the simulations section of the SI uses only M, D, and A.

We now refer to the materials as M, D, and A throughout the manuscript, where necessary.

Figure 2b: The absorption shapes are so closely spaced that the shapes are indistinguishable.

This has been fixed.

Figure 2b: How were the molecules deposited on the glass?

The molecules were deposited by co-evaporation in the UHV as written in the sample fabrication part:

Sample Fabrication: All samples were prepared by thermal vacuum deposition in the UHV ($\leq 10^{-7}$ mbar). Doped samples were fabricated by co-evaporation of the materials, while controlling each deposition rate independently with a respective microbalance. The photoexcited samples were prepared on glass substrates (Borofloat 33, Schott) that were precleaned by acetone and isopropanol treatment in an ultrasonic bath.

Figure 3: How was the Dexter radii determined?

From computer simulations, by estimating the overlap of orbitals participating in the direct charge transfer. Note that the cartoon in Fig. 3 is only an illustration: in our case the Dexter radii are smaller than 1 nm (molecular size) and the Förster radius of 2.4 nm is certainly larger than the Dexter radius.

Page 6, paragraph 2: “singlet of the acceptor” to “singlet excited state of the acceptor”

Fixed.

The simulations assume random orientation ($k_2 = 2/3$) but the calculated J uses $k_2 = 0.476$. Why the discrepancy?

One has to distinguish between random orientation in a dynamic (solution) and static (rigid media) case. $k_2 = 2/3$ is only valid for rapidly reorienting dipoles. For randomly oriented but static/immobile emitters $k_2 = 0.476$ has been derived to be more appropriate by Baumann et al (see Ref. 52 in the SI, stated for the used value).

Figure 4c and 4d: For clarity the y-axis should be labeled Normalized Intensity.

The y-axis label of Figure 4c and 4d has been changed to “Normalized Intensity”.

The order of the SI does not match the order in the text. For the sake of the reader, the organization should coincide.

The order of the SI has been adapted to the order in the text.

What is BA2 from Figure S4?

BA2 is 2,5,8,11-Tetra-*tert*-butylperylene (TBPe). The abbreviation BA2 has been replaced by TBPe and the formula structure was added as an inset.

Figure S10: For those who print in black and white it would be useful to change the shape of one of the circles.

Circles of “no Dexter” have been changed in shape from filled to open circles.

It would be helpful to see the coordinates in the CIE diagram in the SI.

The CIE diagram was added as Fig. S9:

The main text and the SI have been adapted accordingly.

“A comparison of the corresponding CIE-coordinates in Table shows that the emission color of the phosphorescent sensitizer is indeed preserved.”

REVIEWERS' COMMENTS:

Reviewer #3 (Remarks to the Author):

Based on the revisions performed by the authors, I recommend its publication in the present form.

Reviewer #4 (Remarks to the Author):

The authors have addressed most of my comments and the manuscript merits publication almost as is. I have only one additional, but minor follow-up concerns.

Regarding, How was the Dexter radii determined?

Author response: From computer simulations, by estimating the overlap of orbitals participating in the direct charge transfer. Note that the cartoon in Fig. 3 is only an illustration: in our case the Dexter radii are smaller than 1 nm (molecular size) and the Förster radius of 2.4 nm is certainly larger than the Dexter radius.

Reviewer reply: I don't fully understand the note that "the cartoon in Fig. 3 is only an illustration". As in the RFRET and R Dexter are arbitrarily made up? Why? Why not use the calculated values? If not, it should be clearly explained that as drawn, the radii are arbitrary. If calculated values are used than the method for determining the Dexter radii should be included in the experimental details and the methods cited appropriately.

Reviewer comments (black), our responses (blue)

Reviewer #4

I don't fully understand the note that "the cartoon in Fig. 3 is only an illustration". As in the RFRET and R Dexter are arbitrarily made up? Why? Why not use the calculated values? If not, it should be clearly explained that as drawn, the radii are arbitrary. If calculated values are used than the method for determining the Dexter radii should be included in the experimental details and the methods cited appropriately.

The radii presented in the manuscript are of course not arbitrary. The R_{FRET} corresponds to the value calculated from the spectral overlap, while R_{DEXTER} is estimated from MD simulations as an average distance between the neighboring molecules (it is discussed in the Supporting Information). The real Foerster and Dexter rates cannot be depicted by spheres, since they are different for different molecular pairs. They are of the order of the size of a molecule, and hence cannot be sketched to scale in Fig. 3. We now mention in the caption that these quantities are not to scale.